# Effects of Different Grazing Treatments on the Root System of *Stipa krylovii* Steppe

**Tian Tian** [1,2,†], **Jianying Guo** [2,3,†], **Zhenqi Yang** [2,3,*], **Zhenyu Yao** [2,3], **Xinyu Liu** [2,3] **and Ziwei Wang** [1,2]

1    Desert Science and Engineering College, Inner Mongolia Agricultural University, Hohhot 010018, China; tiantian@emails.imau.edu.cn (T.T.); wangziwei123@emails.imau.edu.cn (Z.W.)
2    Yinshanbeilu National Field Research Station of Steppe Eco-Hydrological System, China Institute of Water Resources and Hydropower Research, Beijing 100038, China; guojy@iwhr.com (J.G.); yaozy@iwhr.com (Z.Y.); liuxinyu@iwhr.com (X.L.)
3    Institute of Water Resources for Pastoral Area, Ministry of Water Resources, Hohhot 010020, China
*    Correspondence: yangzq@iwhr.com; Tel.: +86-182-4811-4667
†    These authors contributed equally to this work.

**Abstract:** Plants' root properties are closely related to their ecological adaptability. This study aimed to clarify the differences in root properties of *Stipa krylovii* under different grazing disturbances. The morphological characteristics of root length, root surface area, root volume, root tip number, specific root length, and specific surface area of *S. krylovii* were compared under no grazing, light grazing, moderate grazing and heavy grazing conditions. The ecological adaptability to grazing pressure was also examined. Results showed that the underground biomass density decreased with the increase in grazing intensity. Grazing disturbance can lead to changes in plant community characteristics, and roots adapt to changes in these environmental factors by adjusting their distribution. Among the six root configuration parameters, those under light grazing were significantly higher than those under the other grazing types. The root length and root surface area were concentrated in the range of 0–2 mm. Mild grazing and moderate grazing were conducive to fine root penetration and contact with soil. Moderate grazing disturbance was beneficial to grassland vegetation productivity and played an important role in the stability and sustainable utilization of grassland ecosystem.

**Keywords:** grazing treatment; *Stipa krylovii* steppe; root system

## 1. Introduction

Grassland is one of the largest and most extensive types of terrestrial ecosystems. The global grassland area is estimated to be $3.5 \times 10^9$ hm$^2$, accounting for about 40% of the land area [1]. As a large country for grassland resources, the total area of grassland in China accounts for roughly 41.5% of the land area [2]. Grazing is one of the important measures in grassland management [3]. With the intensification of human activities, grassland grazing disturbance has become a concern [4]. Hence, livestock grazing management remains very important for future grassland protection [5–7]. Studying the influence of grazing on grassland and understanding the processes and mechanisms of grassland degradation under the influence of grazing to take reasonable management measures are of great significance in preventing grassland degradation and ensuring the sustainable development of grassland animal husbandry.

In the grassland ecosystem, grazing grassland constitutes an organic whole composed of herbage–soil–livestock, which interact with and restrict each other. Roots are the "central carrier" of soil and plants, and an important link between plants and soil. They determine the role of grassland vegetation and soil environment [8]. Good root properties improve the utilization efficiency of soil nutrients and water, and the productivity of plant communities [3,4]. Roots intersperse, entangle, and consolidate in the soil, filling the gaps between the soil particles by decomposing the secretions. Enhancing soil organic matter

and aggregating biodegradation improve the soil structure [9]. Grazing is one of the most common types of disturbance affecting grassland communities and structures. Under the trampling and feeding of livestock, the aboveground parts of plants are destroyed, the photosynthetic capacity is weakened, and the vegetation coverage is reduced, thus changing the composition of plant communities. Grazing impacts root growth and production, with different effects at different grazing levels—the higher the grazing intensity, the higher the negative effects on the roots [10]. The growth status of roots is reflected in the soil distribution. Good root architecture improves the efficiency of roots regarding soil nutrient and water use and reflects the influence of the environment on roots.

The *Stipa krylovii* steppe is a transitional zone between desert steppe and typical steppe in northern China and has a mainly temperate continental climate and temperate monsoon climate. The annual precipitation level is low, and the seasonal distribution is uneven, making it prone to floods and other hazards. In recent years, drought has caused serious harm to grasslands, and this effect has intensified with global warming. Grassland desertification and soil erosion are mutually causal and common, forming a vicious circle. *S. krylovii* is important in soil and water conservation, vegetation restoration, and windbreak and sand fixation and maintains ecological stability. The effects of grazing on plant roots are complex [11–13].

Several different views have been established regarding the effect of grazing on plant roots. First, grazing may reduce the growth of plant roots. Livestock changes the effective distribution of biomass by feeding on the aboveground part and slowing down the growth of roots. Root biomass decreases with the increase in grazing intensity [14]. Second, conservative grazing promotes the growth of plant roots and can produce compensatory growth to increase root biomass and net productivity [15]. Third, grazing can have no significant effect on underground roots [16]. Changes in species composition induced by grazing disturbance, mainly those related to a reduction in herbaceous- and total-plant cover or the replacement of herbaceous plants by woody plants, may also lead to variations in the horizontal and vertical distribution of the root biomass in soil [17,18].

At present, most studies on the effects of grazing intensity on plant roots in the desert steppe focus on the analysis of root characteristics of plant communities; only a few explore the differences in root attributes of single plants [15]. Therefore, the current research work explored the response and adaptation of *S. krylovii* to different grazing intensities and explained the ecological strategy, and found the optimal index that can reflect the ecological adaptability of plant roots. This work aimed to provide a scientific basis for the maintenance of grassland community biodiversity and the prevention and control of grassland degradation.

## 2. Materials and Methods

### 2.1. Study Area

The desert grassland ecological/hydrological field scientific observation and research station at the northern foot of Yinshan Mountain was selected as the sample plot. The area is administratively located in Xilamuren Town, Darhan Maoming and United Banner, Baotou City, Inner Mongolia, with geographical coordinates of 41°12′–41°59′ N, 110°25′–111°27′ E (Figure 1). The climate is a semi-arid and temperate continental climate, cold and dry in winter, and with a short and warm summer. The interannual variation is large, and the precipitation is mainly concentrated in July–September in summer. The soil in the test area is chestnut soil, the thickness of the soil layer is approximately 40 cm, and the lower part is a calcic layer. The vegetation communities are mainly *Stipa breviflora*, *S. krylovii*, *Leymus chinensis*, and *Agropyron cristatum*.

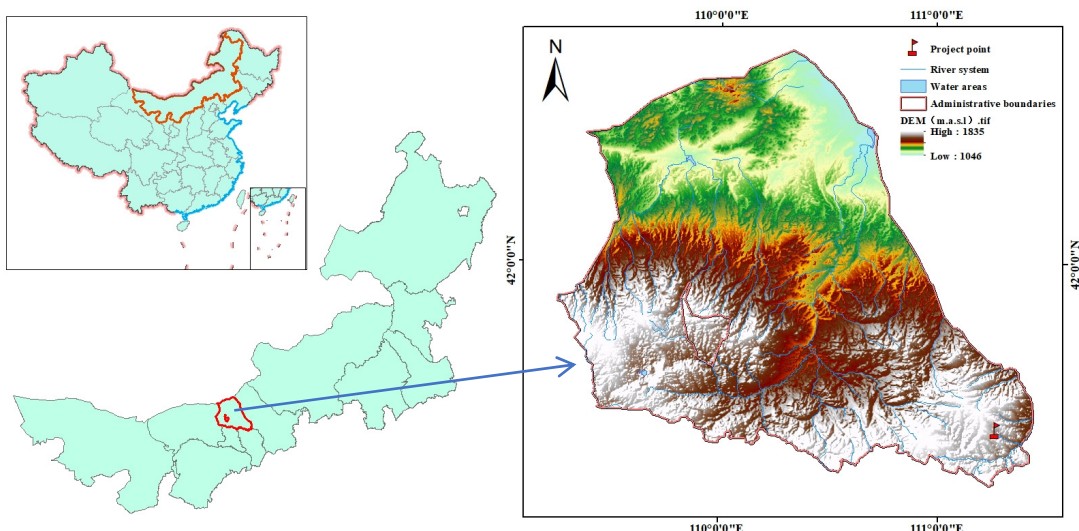

**Figure 1.** Overview of the study area.

### 2.2. Plot Settings

The grazing experimental area of the ecological/hydrological field scientific observation and research station of the desert steppe in the northern foot of Yinshan Mountain was selected for this study. The vegetation species composition, forage yield, and litter quantity of the grassland were investigated and analyzed using the line transect method in late July 2018 according to the standard of livestock-carrying capacity of the Chinese agricultural industry and previous research on the livestock-carrying capacity of the desert steppe in northern China [19,20]. A completely randomized block design was adopted, and the grazing land was divided into heavy grazing (HG), moderate grazing (MG), light grazing (LG), and no grazing (CK). According to the classification criteria, 100 m long and 100 m wide grazing plots were established on slopes with angles of 3°–6°. The fenced grazing area was divided into three blocks with three repetitions for a total of 12 plots. Each plot was fenced with a 1.5 m high wire mesh fence and equipped with small doors to control the number of sheep. The grazing conditions were HG (3 sheep/hm$^2$), MG (2 sheep/hm$^2$), LG (1 sheep/hm$^2$), and CK (0 sheep/hm$^2$). The sheep used were adult sheep with a similar body size [21]. The grazing experiment commenced in May and concluded at the end of November. To ensure the representativeness and accuracy of the research methodology, we adopted the same grazing system as that employed by local herdsmen. Additionally, a feeding schedule was implemented for the sheep from 7 a.m. to 7 p.m., followed by their return to the sheep pen for rest during the nighttime.

### 2.3. Sample Collection and Processing

The collection of root samples for this study took place in July 2023. In each plot, 10 herbaceous plants with good growth were selected. Root–soil complex samples with a length and width of 15 cm and a depth of 20 cm were collected with the base of the selected plants (constructive species) as the center, packed into a self-sealing bag, brought back to the laboratory, and stored in a low-temperature refrigerator at 4 °C. The samples were repeatedly washed with a 0.5 mm sieve in the room until all the roots had been washed out. The roots of the plants were preserved at low temperature and brought back to the room for cleaning. A small amount of distilled water was poured into the root disc, and the roots were tiled and separated to ensure dispersion without crossing. The root parameters were scanned using a 11000XL scanner (Epson, Tokyo, Japan) at 1200 dpi, and the root images were analyzed using WinRHIZO Pro 2012b software (Regent Instruments Inc., Quebec City, QC, Canada) to measure root length, root diameter, root surface area, root volume, and other parameters. After scanning, the roots were dried in an oven (60 °C, 48 h) to determine the drying quality of the roots.

*2.4. Data Processing*

The specific root length and specific surface area were calculated based on the obtained root parameters [22].

$$SRL = RL/RB, \tag{1}$$

$$SSA = RSA/RB \tag{2}$$

In the formula, SRL means specific root length (cm/g), SSA means specific surface area ($cm^2$/g), RL means root length (cm), RSA means root surface area ($cm^2$), and RB means root biomass.

Excel 2013 was used for data collation, and SPSS 26 for data analysis. One-way analysis of variance was applied to analyze significant differences between the root characteristic indexes of different grazing intensities, and Tukey's HSD was adopted for multiple comparisons. The significance level was set to 0.05. Correlation analysis and principal component analysis were performed on root-characteristic indexes. Origin 2022 was used for mapping, and significant markers were added.

## 3. Results

*3.1. Differences in Root Biomass Density under Different Grazing Intensities*

The aboveground biomass of *S. krylovii* decreased with the increase in grazing intensity (Figure 2a). Compared with that under CK, the aboveground biomass of *S. krylovii* decreased by 37.39%, 73.30%, and 82.60% under LG, MG, and HG, respectively.

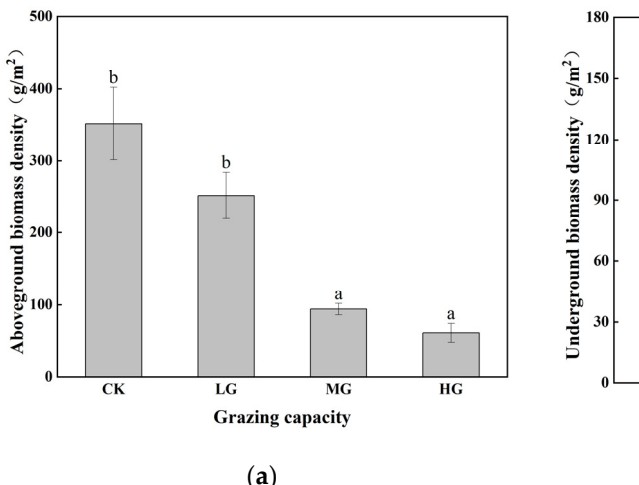
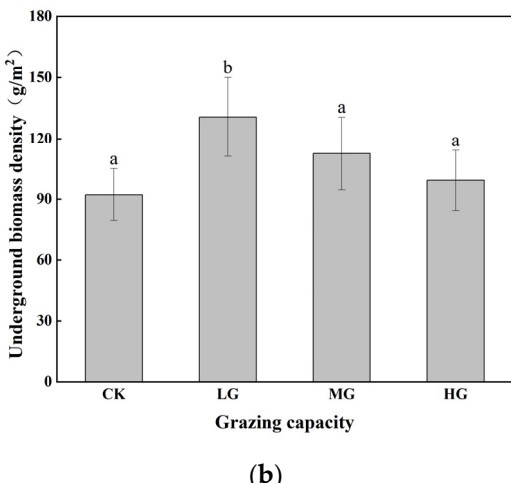

(**a**)                                                                                   (**b**)

**Figure 2.** Changes in *Stipa krylovii* biomass density under different grazing intensities: (**a**) aboveground and (**b**) underground. Letters indicate differences among different grazing intensity groups for the same indicator (*p* < 0.05). Significant differences between variables are marked by different letters.

The underground biomass density gradually decreased with the increase in grazing intensity (Figure 2b). Underground biomass density was maximized under light grazing. Underground biomass density in the other treatments was not significantly different, but decreased by 6.98%, 22.76%, and 33.06% compared to LG. These results show that LG promoted root growth.

*3.2. Differences in Root Distribution in Vertical Space under Different Grazing Intensities*

A significant positive correlation between many root parameters of *S. krylovii* was determined (Table 1). The root surface area and root volume were significantly positively correlated with the specific root length. The root volume was positively correlated with the root surface area, root tip number, and specific root length but negatively correlated with the root tip number. The number of root tips was positively correlated with the specific root length. A significant positive correlation was found between specific root length and specific surface area (Table 1).

**Table 1.** Correlation coefficient of *Stipa krylovii* root attributes.

| Vegetation | | Root Length | Root Surface Area | Root Volume | Root Tips | Specific Root Length |
|---|---|---|---|---|---|---|
| *Stipa krylovii* | Root surface area | 0.872 ** | | | | |
| | Root volume | 0.588 ** | 0.873 ** | | | |
| | Root tips | 0.913 ** | 0.788 ** | 0.547 ** | | |
| | Specific root length | 0.261 | 0.018 | −0.105 | 0.325 * | |
| | Specific root surface area | 0.221 | 0.191 | 0.179 | −0.280 | 0.886 ** |

Note: * indicate differences among different grazing intensity groups for the same indicator (*p* < 0.05), and ** indicate differences among different grazing intensity groups for the same indicator (*p* < 0.01). Significant differences between variables are marked by different * or **.

The changes in the six root indexes of S. krylovii under different grazing intensities differed (Figure 3). The order of root length was LG > MG > HG > CK (Figure 3a). The root length of the LG sample was significantly different from that of the other grazing samples (*p* < 0.05). Compared to the CK, grazing significantly enhanced the vegetation root length; however, an inverse relationship between grazing intensity and root length was observed. In particular, conservative grazing increased the root length of the vegetation. For root surface area and root volume (Figure 3b,c), the change rule was consistent. The root surface area and root volume under LG and MG were significantly higher than those under HG and CK (*p* < 0.01), but no significant difference was observed between HG and CK.

The number of root tips (Figure 3d) in the LG plot was also significantly higher than that in other grazing plots. The order of specific root length was CK> HG> LG > MG (Figure 3e), and no significant difference in the specific surface area of *S. krylovii* was observed under different grazing intensities (Figure 3f). Overall, LG was beneficial to promote the longitudinal growth of roots.

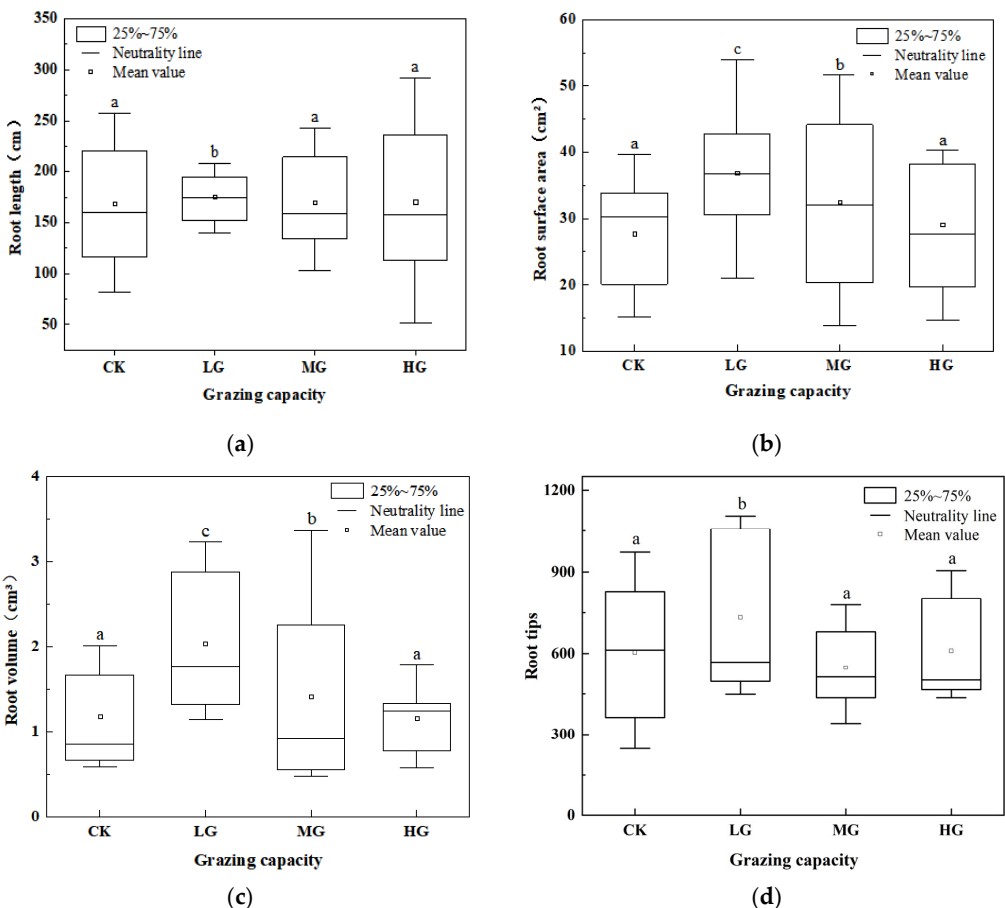

**Figure 3.** *Cont.*

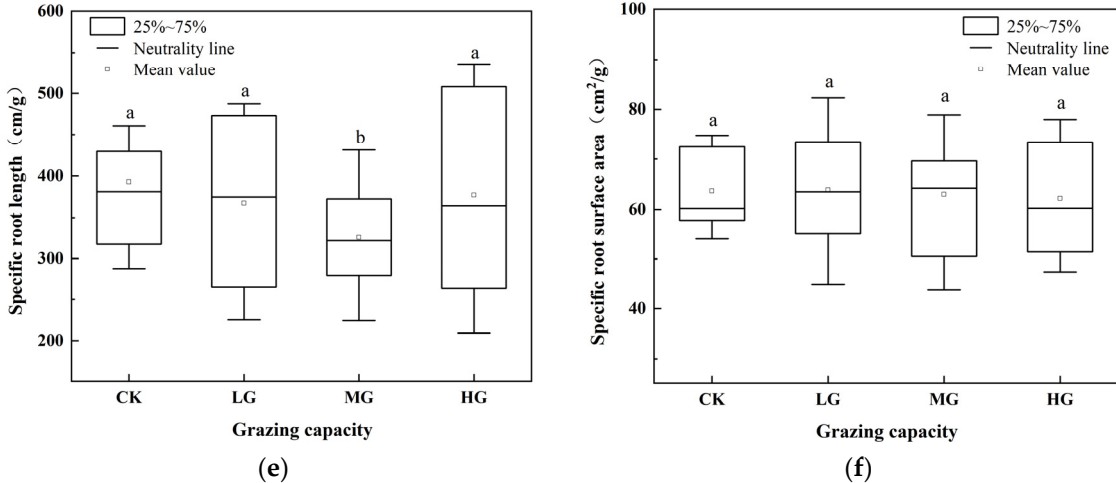

**Figure 3.** Spatial variation of *Stipa krylovii* root properties under different grazing intensities: (**a**) root length, (**b**) root surface area, (**c**) root volume, (**d**) root tips, (**e**) specific root length, and (**f**) specific root area. Note: Letters indicate differences among different grazing intensity groups for the same indicator ($p < 0.05$). Significant differences between variables are marked by different letters.

### 3.3. Differences in Diameter Class Distribution of Stipa krylovii under Different Grazing Intensities

The composition of various diameter classes was represented by the length, surface area, and volume of roots in different diameter classes under varying grazing intensities (Figure 4). The significance of root length, root surface area, and root volume of the same diameter class among single vegetation types under different grazing intensities exhibited certain rules. The root diameter was divided into five grades, with 1 mm as the gap. In the range of the 0–1 mm diameter class, the order of root length was LG > MG > CK > HG and the order of root surface area was MG > LG > CK > HG. The root volume under different grazing intensities was not different, and that under HG was the highest. In the range of 1–2 mm diameter, the order of root length, root surface area, and root volume consistently showed the order of LG > MG > HG > CK. In the range of 2–3 mm diameter class, the order of root length was consistent with that in the 0–1 mm diameter class and the order of root surface area and root volume was consistent with that in the 1–2 mm diameter class. In the range of the 3–4 mm diameter class, the root length, root surface area, and root volume also showed the order of LG > MG > HG > CK. When the diameter was >4 mm, the root length was LG > HG > MG > CK, the root surface area was LG > MG > HG > CK, and the root volume was MG > LG > HG > CK.

Root length and root surface area were concentrated in the range of 0–2 mm with an average proportion of more than 85%. Compared with that in the other plots, the content of fine roots in the CK plots was significantly lower and that in the LG and MG plots was generally higher. The external interference of livestock significantly increased the content of underground fine roots, and its diameter class composition also changed significantly.

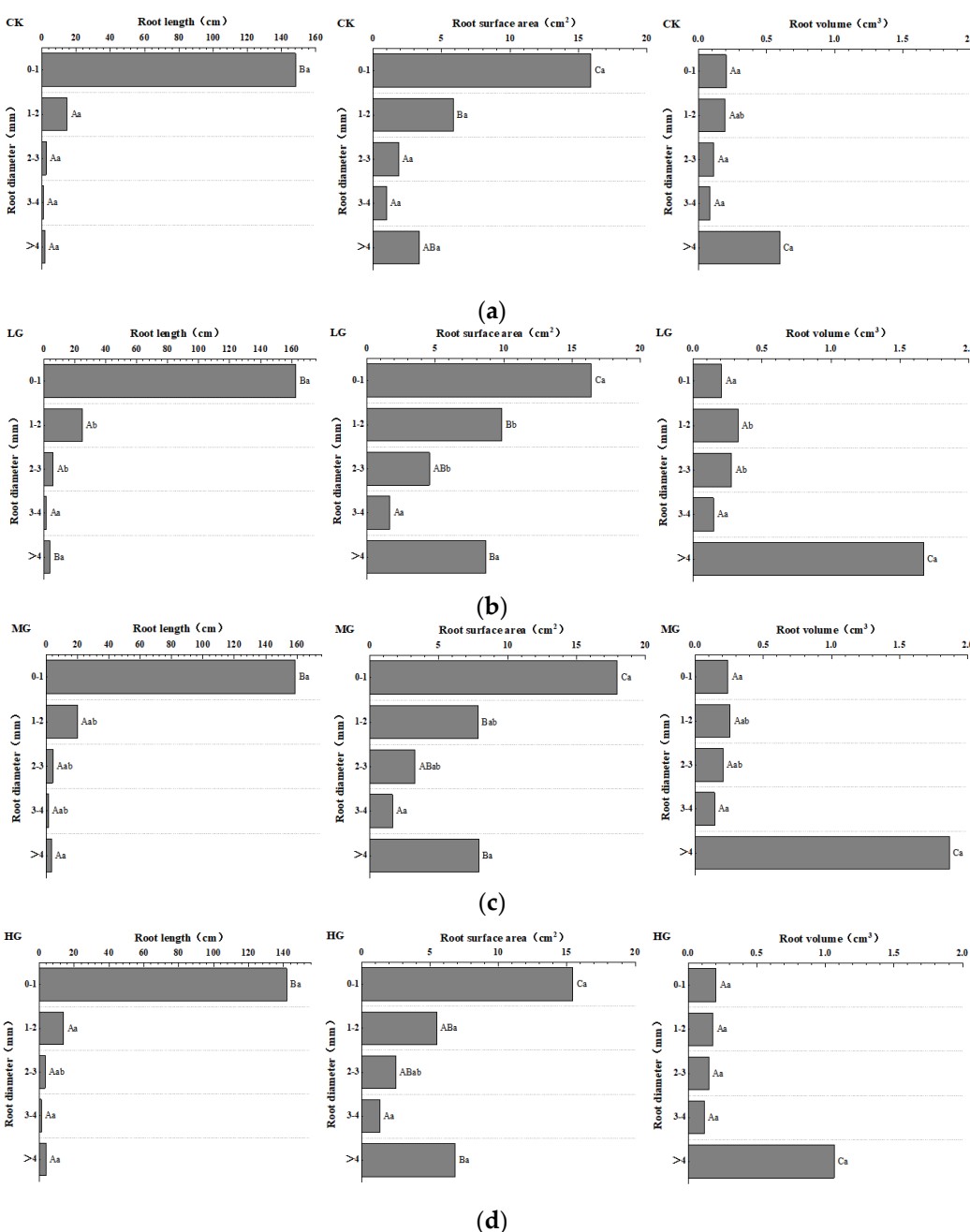

**Figure 4.** Distribution of root length, root surface area, and root volume of *Stipa krylovii* under different grazing intensities: (**a**) no grazing, (**b**) light gazing, (**c**) moderate grazing, and (**d**) heavy grazing. Note: Small letters illustrate differences among multiple grazing intensity groups for the same soil depth ($p < 0.05$). Capital letters represent differences among varying soil depths for the same grazing intensity ($p < 0.05$). Significant differences between variables are marked with different letters.

### 3.4. Differences in Ecological Adaptability of Stipa krylovii under Different Grazing Intensities

The important root parameters of *S. krylovii* were extracted by principal component analysis (Table 2). The KMO of *S. krylovii* was calculated to be 0.485, and the Bartlett sphericity test reached a significant level ($p < 0.01$). The variation was 57.70% and 29.97%, and the cumulative was 87.67% (>85%). According to the component matrix of *S. krylovii*, the feature vector (Table 3) was calculated and the component expression of *S. krylovii* was obtained. According to the contribution rate of the principal component and the coefficients corresponding to the root parameters in the principal component, the comprehensive score Y of the principal component was calculated.

**Table 2.** Principal component analysis of *Stipa krylovii* root attributes.

| Vegetation | Principal Component | Total | Variability (%) | Cumulative (%) |
|---|---|---|---|---|
| *Stipa krylovii* | 1 | 3.462 | 57.70 | 57.70 |
|  | 2 | 1.798 | 29.97 | 87.67 |

**Table 3.** Principal component eigenvector matrix of *Stipa krylovii* root attributes.

| Vegetation | Principal Component | Root Length | Root Surface Area | Root Volume | Root Tips | Specific Root Length | Specific Root Surface Area |
|---|---|---|---|---|---|---|---|
| *Stipa krylovii* | 1 | 0.498 | 0.499 | 0.414 | 0.489 | 0.189 | 0.237 |
|  | 2 | −0.078 | −0.236 | −0.276 | −0.013 | 0.690 | 0.621 |

High scores of root length, root surface area, and root tip number in Y1 (Formula (3)) and high scores of specific root length and specific surface area in Y2 (Formula (4)) indicate that principal component 1 and principal component 2 reflect all the information of *S. krylovii* root attributes. The final relationship between Y1 and Y2 was obtained from Y (Formula (5)), which is the main parameter affecting the root characteristics of *S. krylovii* and is used to explain the ecological adaptability of root morphology.

$$Y1 = 0.498x_1 + 0.499x_2 + 0.414x_3 + 0.489x_4 + 0.189x_5 + 0.237x_6, \tag{3}$$

$$Y2 = 0.078x_1 - 0.236x_2 - 0.276x_3 - 0.013x_4 + 0.690x_5 + 0.621x_6 \tag{4}$$

$$Y3 = 0.5770Y1 + 0.2997Y2 \tag{5}$$

## 4. Discussion

The feeding practices of domestic animals during grazing have a direct and significant impact on the composition and biomass of grassland vegetation communities [23]. As grazing intensity increases, the aboveground biomass of the desert steppe in Inner Mongolia exhibits a declining trend, which aligns with Ren et al.'s findings [24]. Grazing exerts both short-term and long-term effects on grasslands, with leaves being the primary target for animal consumption. By consuming plant leaves, animals influence the utilization of photosynthetic substances by plants, stimulating compensatory growth and consequently altering underground plant biomass [25]. This leads to an increased proportion of underground biomass relative to above-ground biomass. The results from different grazing intensities demonstrate that higher grazing pressure does not result in a decrease in underground biomass over 6 years compared to non-grazing plots; instead, light grazing significantly enhances underground biomass—a trend consistent with the Xilin River observations [26]. Furthermore, the above-ground portion of plants plays a crucial role in supporting root system development by providing necessary energy sources. In turn, the root system affects above-ground growth through water and nutrient absorption—this is the primary factor influencing vegetation's underground biomass [27]. The observation made by Ricardo Henrique Ribeiro indicates that grazing induces alterations in root systems, irrespective of the duration of grazing [28]. Although light, moderate, and heavy grazing all contribute to an increase in root biomass, it is crucial to consider the potential long-term detrimental effects associated with heavy grazing. In comparison to medium- and low-intensity levels, heavy grazing results in elevated soil bulk density and compaction [29,30], reduced soil carbon storage capacity [31], and decreased system productivity [32]. These factors are not conducive to the long-term recovery of grassland vegetation productivity. However, it should be noted that the impact of grazing on plant roots may vary depending on the specific plant species. Hoogsteen et al., through a simulated grazing experiment conducted in the Netherlands, discovered no significant effect of grazing on root biomass across any soil layer [33].

The compensatory regeneration of plants following grazing is crucial for the restoration of plant production and nutrient stock [34]. The compensatory regeneration of plants following grazing also plays a crucial role in the restoration of plant productivity and nutrient reserves [35]. The uptake of nutrients by grassland plants contributes to 32% of nitrogen and 28% of phosphorus [36], while grazing significantly reduces plant biomass, thereby increasing the reliance on soil root nutrient uptake for plant nutrient conservation [37]. Significant correlations were observed among the root properties of Stipa clanneri, indicating a certain interdependence between these parameters. Principal component analysis revealed that key factors influencing the root system included root length, root surface area, root volume, and root tip number. Both root length and surface area play vital roles in determining *Stipa kirstii*'s characteristics as a densely clustered plant with limited water storage capacity [38]. However, due to adventitious roots formed by tillers, well-developed roots facilitate the efficient absorption of water and nutrients while promoting the growth of *Stipa kirstii* [39]. Studies on perennial pastures have demonstrated that conservative grazing induces changes in root morphology by stimulating the formation of new and finer roots, thereby enhancing nutrient and water uptake [40]. Among the six root architecture parameters, further classification revealed that within the 0–2 mm diameter class, the root length of lightly grazed and moderately grazed plants was significantly greater than that of heavily grazed and non-grazed plants. Additionally, the root surface area of moderately grazed plants was significantly higher compared to lightly grazed and heavily grazed plants. This is consistent with the results obtained by Pucheta et al. in Pampa de Achala, who found that grazing will increase the content of fine roots [41]. The impact of grazing on root growth is evident, as it stimulates the development of fine roots and alters the distribution of root diameter classes to adapt to the grazing environment. This adaptation ensures the efficient absorption of water and nutrients [42].

## 5. Conclusions

After 6 years of grazing treatment, it was observed that grazing significantly influenced the vegetation biomass in a desert grassland area. As grazing intensity increased, both aboveground and underground biomass densities gradually decreased, although the underground biomass density was higher under grazing treatment compared to the control. Grazing promoted the development and growth of underground roots as well as enhancing vegetation resistance to growth. The specific surface area did not differ among the six root architecture parameters of *Stipa krylovii*. Under control conditions, the specific root length was higher compared to other grazing treatments. The root length, root surface area, root volume, and root tip number were higher under light grazing compared to other grazing treatments. Based on the grading of root diameter classes, both root length and surface area were concentrated within the range of 0 mm < d ≤ 2 mm; light and moderate grazing were more favorable for fine roots' penetration and contact. These findings demonstrate that conservative grazing plays a pivotal role in enhancing grassland vegetation productivity and ensuring the stability and sustainable utilization of the grassland ecosystem. Consequently, it is imperative to implement scientifically sound grazing strategies in grassland management to uphold the vitality and equilibrium of the grassland ecosystem.

**Author Contributions:** Data processing, writing—original draft preparation, and editing, T.T. and J.G.; manuscript validation and revision, Z.Y. (Zhenqi Yang); project administration and funding acquisition, Z.Y. (Zhenyu Yao); data collection, X.L. and Z.W.; T.T. and J.G. are co-first authors. All authors have read and agreed to the published version of the manuscript.

**Funding:** This study was supported by the National Natural Science Foundation of China (No. 42177347), IWHR Research and Development Support Program (MK2022J1D), The influence mechanism of grassland patch pattern on the hydraulic erosion process of grazing grassland (MK2022J11), IWHR Research and Development Support Program (MK2023J06) and IWHR Research and Development Support Program (MK2023J05).

**Institutional Review Board Statement:** Not applicable.

**Informed Consent Statement:** Not applicable.

**Data Availability Statement:** The unavailability of data is attributed to privacy or ethical constraints.

**Acknowledgments:** We would like to thank all the partners involved in this study and the experimental platform provided by the Yinshanbeilu National Field Research Station of Steppe Eco-Hydrological System. The authors would like to thank the reviewers for their helpful and constructive comments and suggestions which have substantially improved the quality of this manuscript.

**Conflicts of Interest:** The authors declare no conflicts of interest.

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
