# Peer review of "Effects of Different Grazing Treatments on the Root System of Stipa krylovii Steppe"

_sustainability, doi:10.3390/su16103975_

Round 1

Reviewer 1 Report

Comments and Suggestions for Authors

The manuscript involves an interesting study of grass root properties following grazing treatments. Some necessary information is missing, mostly about the specifics of the grazing treatment and the period of time between grazing and plant sampling. The results are sometimes not presented clearly, and more attention needs to be paid to which results are significant and accurately reflecting that in the text. For example, you show many properties were maximized under light grazing, but many properties are summarized as having a negative correlation with grazing intensity. This ignores this difference between light grazing and no grazing. The discussion could be improved to more clearly discuss your results and how they fit in with other studies. Lastly, because this study involves grazing, more discussion of what these results would mean for grazing and how similar the grazing treatments are to real world grazing is needed.

Line 46: Remove "the ecological environment,". This is vague

Line 48: Connect these two sentences with a comma

Lines 53-57: This information is a little vague and repetitive, and needs a citation. I recommend changing the sentence to something like "Grazing impacts root growth and production, with different effects at different grazing levels (CITATION)"

Line 61: Change "annual precipitation is small" to "annual precipitation is low"

Line 66-67: This needs a citation, and the statement about "forming a vicious cycle" is strong language you may want to change to something like "creating a positive feedback"

Line 69: This should be the start of a new paragraph

Line 69-70: To make this more clear, I recommend changing this to "First, grazing may reduce the growth of plant roots"

Line 73: What does "MG" mean? Is this supposed to say "grazing"?

Line 73: Again, I would change this to "grazing can reduce..."

Line 75: Change to "grazing can have no significant effect"

Line 79: Start new paragraph here

Line 85: Remove the comma and space after "morphology"

Lines 60-67: This information about S. krylovii steppe is better in the Study Area section, and is already mostly covered there. I recommend removing this information from the introduction, and adding more specifics about annual precipitation and temperature to the Study Area section.

Lines 102-115: There is important information missing from the study design. What time of year was grazing applied? How long we sheep allowed to graze? What was the time period between grazing and your sampling?

Lines 132-133: These unit definitions need to be provided either within the text in a complete sentence, or as formulas and directed to within the text

Lines 142 and 147: It is more clear to say "The aboveground biomass of S. krylovii decreased with the increase in grazing intensity (Fig 2a)" rather than starting the sentence with "As shown in Fig.2-a"

Lines 145-147: This sentence should not be in the Results section

Line 147: You should note that only LG was significantly different than the other treatments. Therefore, I think this result should be summarized as "Underground biomass density was maximized under light grazing. Underground biomass density in the other treatments was not significantly different than one another, but were deceased by _%, _%, and _% compared to LG"

Line 157: Not all the correlations shown in Table 1 are positive. Change this to "Table 1 shows significant positive correlations between many root parameters..." 

Line 170-173: Start by noting that only LG is significantly different than the others. In the boxplot it is not possible to see any difference in median between the other treatments. The sentence "In particular, MG increased the root length of vegetation" is misleading and not correct, and would require you to provide the median or mean values for that treatment and others

Line 172: "The root length showed a downward trend with the increase in grazing intensity" Your findings do not really reflect this, because CK has the minimum root length. This needs to be made more clear.

Line 182: What statistical test was used to test differences between groups? You say P-Values were used, but from what test?

Line 229: This should be Line 4, not Line 3.

Line 229-237: I don't understand what is trying to be shown here. The equations need to be given more informative text rather than "This is example _ of an equation"

Line 258: I think you should say what study found this by name

Line 261-263: I don't think root length ratio or surface area ratio were shown in the results section, they should be

Discussion: There is a lot of information here and very long paragraphs. I think you can make it more clear what results you are talking about and how they fit into other research. Right now a lot of the information from Lines 273-295 looks more like Introductory information

Discussion and Conclusion: The differences between CK and LG are interesting and I don't think you discuss them enough.

Discussion: How similar are the grazing levels you applied to typical grazing in the area? Is grazing commonly closer to the HG level, MG, LG? This greatly shapes how to interpret your results. Knowing the period of time between grazing and measurement is also very important. How long lasting are the effects of grazing on root properties and other plant properties?

Reviewer 2 Report

Comments and Suggestions for Authors

The authors studied the root characteristics of steppe grasses in which they combined extensive detailed field work with modern technologies. Introduction is well written and methodology are is presented in a way which allows the replication of the experiment. The only significant mistake is the wrong level of significance of 0.01 which is mentioned there. Results are well presented by the usage of interesting graphs, but the text needs some technical polishing. Some paragraphs are located in wrong sections (i.e. discussion in results). The manuscript is a little wordy in some parts for no reason, which can be easily corrected. The biggest problem is the discussion. Many statements aren’t backed up with references. A large part of discussion is just listing statements without any context or references. The discussion needs to be improved significantly by comparing your own results, and your and other researchers’ results, and drawing conclusions from that, not only by listing your results and then listing other results. All the suggestions for changes are listed in the attached file.

Comments on the Quality of English Language

The quality of the English language is satisfactory, and needs only minor changes.

Round 2

Reviewer 1 Report

Comments and Suggestions for Authors

The manuscript is significantly improved in this revised draft and I appreciate the authors' attention to prior comments. I have no further editing comments on the manuscript at this point, and found the experimental design and results much easier to interpret in this manuscript.

Author Response

The text has been revised in accordance with the grammatical and tense issues pointed out by the reviewer.

Reviewer 2 Report

Comments and Suggestions for Authors

Dear authors,

You have significantly improved the manuscript. There are only a couple of technical correction, and I agree to accept it.

Comments on the Quality of English Language

Author Response

(The authors gave the same response as above.)
